# Endothelial Glycocalyx Integrity in Treatment-Naïve People Living with HIV before and One Year after Antiretroviral Treatment Initiation

**DOI:** 10.3390/v15071505

**Published:** 2023-07-05

**Authors:** Paraskevi C. Fragkou, Ignatios Ikonomidis, Dimitrios Benas, Dimitra Kavatha, Charalampos D. Moschopoulos, Konstantinos Protopapas, Gavriella Kostelli, John Thymis, Dionysia Mpirmpa, Irene Galani, Maria Tsakona, Chrysanthi Oikonomopoulou, George Theocharous, Vassilis G. Gorgoulis, Parisis Gallos, Sotirios Tsiodras, Anastasia Antoniadou, Antonios Papadopoulos, Helen Triantafyllidi

**Affiliations:** 1First Department of Critical Care and Pulmonary Services, Evangelismos Hospital, Athens Medical School, National and Kapodistrian University of Athens, 10676 Athens, Greece; 2Second Department of Cardiology, Attikon University Hospital, Athens Medical School, National and Kapodistrian University of Athens, 12462 Athens, Greece; ignoik@gmail.com (I.I.); dimitriosbenas@gmail.com (D.B.); kosteligavriela@hotmail.com (G.K.); johnythg@gmail.com (J.T.); dbirba@gmail.com (D.M.); seliani@hotmail.com (H.T.); 3Fourth Department of Internal Medicine, Attikon University Hospital, Athens Medical School, National and Kapodistrian University of Athens, 12462 Athens, Greece; dimitra.kavatha@gmail.com (D.K.); bmosxop@yahoo.gr (C.D.M.); kprotopapas@hotmail.com (K.P.); egalani@med.uoa.gr (I.G.); mariatsakwna1@gmail.com (M.T.); xrysanthi_oikonomopoulou@yahoo.gr (C.O.); sotirios.tsiodras@gmail.com (S.T.); ananto@med.uoa.gr (A.A.); antpapa1@otenet.gr (A.P.); 4Molecular Carcinogenesis Group, Department of Histology and Embryology, Athens Medical School, National and Kapodistrian University of Athens, 11527 Athens, Greece; theocharousgiorgos@gmail.com (G.T.); vgorg@med.uoa.gr (V.G.G.); 5Computational Biomedicine Laboratory, Department of Digital Systems, University of Piraeus, 18536 Piraeus, Greece; parisgallos@yahoo.com

**Keywords:** endothelial glycocalyx, HIV, permeable boundary region, antiretroviral treatment, biomarkers, atherosclerosis, integrase inhibitor, protease inhibitor

## Abstract

Endothelial glycocalyx (EG) derangement has been associated with cardiovascular disease (CVD). Studies on EG integrity among people living with HIV (PLWH), are lacking. We conducted a prospective cohort study among treatment-naïve PLWH who received emtricitabine/tenofovir alafenamide, combined with either an integrase strand transfer inhibitor (INSTI, dolutegravir, raltegravir or elvitegravir/cobicistat), or a protease inhibitor (PI, darunavir/cobicistat). We assessed EG at baseline, 24 (±4) and 48 (±4) weeks, by measuring the perfused boundary region (PBR, inversely proportional to EG thickness), in sublingual microvessels. In total, 66 consecutive PLWH (60 (90.9%) males) with a median age (interquartile range, IQR) of 37 (12) years, were enrolled. In total, 40(60.6%) received INSTI-based regimens. The mean (standard deviation) PBR decreased significantly from 2.17 (0.29) μm at baseline to 2.04 (0.26) μm (*p* = 0.019), and then to 1.93 (0.3) μm (*p* < 0.0001) at 24 (±4) and 48 (±4) weeks, respectively. PBR did not differ among treatment groups. PLWH on INSTIs had a significant PBR reduction at 48 (±4) weeks. Smokers and PLWH with low levels of viremia experienced the greatest PBR reduction. This study is the first to report the benefit of antiretroviral treatment on EG improvement in treatment-naïve PLWH and depicts a potential bedside biomarker and therapeutic target for CVD in PLWH.

## 1. Introduction

Despite the advent of potent and well-tolerated antiretrovirals, the human immunodeficiency virus (HIV) epidemic continues to be a major healthcare challenge, posing a significant socioeconomic burden for many countries around the world. It is estimated that 38.4 million people were living with HIV (PLWH) in 2021 [1]. The advances of HIV treatment and, especially, the modern combination antiretroviral therapy (cART) have revolutionized HIV treatment and have drastically decreased mortality and morbidity of PLWH. Over recent decades, HIV has been gradually transformed from a deadly infection to a chronic disease. In fact, acquired immunodeficiency syndrome (AIDS)-related deaths have been reduced by 68% since 2004 and by 52% since 2010 [1]. However, the increase in life expectancy has led to a progressively ageing HIV population which, in turn, faces an increasingly higher incidence of non-communicable diseases and age-related comorbidities that were, until recently, uncommon for PLWH, such as neurodegenerative disorders and cardiovascular disease (CVD).

CVD is among the leading causes of mortality and morbidity among PLWH and HIV infection is now recognized as a major cardiovascular risk factor [2,3]. In fact, PLWH have a 2-fold higher risk of CVD and a 1.5–2-fold higher risk of acute myocardial infarction compared to the general population [4,5,6]. Moreover, CVD-associated mortality among PLWH increased significantly between 1999 and 2013, albeit overall HIV-related mortality reduced during the same period [7]. These data point toward the significance/urgency of establishing an optimal framework for cardiovascular (risk) assessment in the HIV population, to reduce CVD-related mortality and morbidity. However, risk assessment tools and diagnostic approaches specific to PLWH remain remarkably limited, and the ones applicable to the general population are less accurate [8].

The exact pathogenesis of CVD in HIV infection is still unknown, but it is unambiguously a multi-factorial process which encompasses traditional risk factors (such as arterial hypertension, dyslipidemia, diabetes, smoking, etc.) and HIV-related factors, such as specific classes of antiretrovirals, the HIV-induced persistent immune and coagulation activation and microbial translocation [5]. However, the underlying common process is chronic inflammation, which represents a significant stimulus of atherosclerosis development among PLWH [9,10].

Vascular endothelium is a key target of several stimuli (including chronic inflammation) that eventually lead to atherosclerosis and CVD development [11]. Endothelial glycocalyx (EG) is a network of membrane-bound proteoglycans and glycoproteins, along with endothelium- and plasma-derived soluble proteins that cover the luminal surface of endothelium [12]. EG is an essential component of endothelium, and exerts pleiotropic effects on vascular homeostasis, like signaling, mechanotransduction, protection of endothelial cells, and regulation of the vascular barrier [12]. Data from micro-and macro-circulation studies show that several (acute and chronic) conditions, such as sepsis, ischemia and reperfusion, hypertension, hyperglycemia and diabetes, and acute and chronic renal dysfunction lead to EG disruption by reducing its thickness and deforming its structural components [13]. Importantly, EG thickness can be measured easily with non-invasive bed-side methods [14,15].

To our knowledge, there are currently no available studies assessing the integrity of EG in the HIV population within the context of CVD assessment. Based on the premise that all these inflammatory processes are associated with the breach of EG, we hypothesized that HIV infection can/could also lead to EG degradation. EG disruption, in turn, will induce endothelial dysfunction which signifies the first step of the pathway that leads to atherosclerosis and CVD development and may be an early indicator of cardiovascular impairment in PLWH. We also hypothesized that cART initiation and the associated reduction in viral replication, would restore EG integrity. Hence, loss of EG integrity may be an early indicator of cardiovascular impairment in PLWH.

The aim of this study is to evaluate the EG integrity before and after cART initiation and to investigate any potential associations between EG thickness and clinical parameters, including different cART regimens and inflammatory biomarkers.

## 2. Materials and Methods

### 2.1. Study Design and Participant Selection

We conducted a prospective observational study in persons with confirmed HIV infection in Attikon University Hospital, Athens, Greece between December 2017 and March 2020. PLWH were included in the evaluation following written informed consent. Inclusion criteria were age ≥ 18 years old, no prior exposure to cART and initiation of an eligible cART regimen was selected by the treating physician. Eligible cART regimens were emtricitabine (FTC)/tenofovir alafenamide (TAF) as a backbone, combined with either an integrase strand transfer inhibitor (INSTI), (i.e., dolutegravir (DTG), raltegravir (RAL) and elvitegravir/cobicistat (EVG/c)), or with cobicistat boosted darunavir (DRV/c). These regimens were selected based on the recommendations of the European AIDS Clinical Society for initial cART schemes in treatment-naïve PLWH [16,17,18].

Exclusion criteria included poorly controlled diabetes or hemoglobin A1c (HbA1c) ≥ 7.5% before cART initiation, poorly controlled hypertension and/or dyslipidemia requiring initiation or modification of medication before enrolment to the study, morbid obesity (body mass index, BMI > 40), and initiation of a non-eligible cART regimen.

### 2.2. Scheduled Visits

Visits were scheduled at three time points, including before cART initiation (baseline/visit 1), at 24 (±4) weeks (visit 2), and at 48 (±4) weeks (visit 3). Clinical data (such as body weight, BMI, and smoking history) collection, blood sampling for routine tests and assessment of endothelial glycocalyx thickness were performed in all three time points. However, blood for biomarker measurement was drawn only during the first and last visits.

### 2.3. Sample Collection

Blood samples were collected in BD Vacutainer^®^ Ethylenediaminetetraacetic Acid (EDTA) tubes and one BD Vacutainer^®^ Serum Tube, (Becton Dickinson, Oxford Science Park, Oxford, UK). The serum tube was allowed to clot for a minimum of 15 min. One EDTA tube was centrifuged at 3000 round per minutes (rpm) for 10 min at 20 °C, and plasma was subsequently aliquoted in Eppendorf tubes and stored at −80°C until further biomarker analysis. The second EDTA and the serum tubes were sent to the hospital laboratory for routine tests.

### 2.4. Routine Blood Tests

On visits 1, 2, and 3, routine blood tests were performed, including full blood count, high density lipoprotein cholesterol (HDL-c), low-density lipoprotein cholesterol (LDL-c), total cholesterol, triglycerides, creatinine, liver function tests, CD4+ T-lymphocytes, CD4+/CD8+ lymphocyte ratio, and viral load. Viral loads <50 copies/mL were considered undetectable.

### 2.5. Biomarkers Measurement

We measured the levels of high sensitivity C-reactive protein (hsCRP), d-dimers, and interleukin (IL)-6 using commercially available Enzyme-Linked Immunosorbent Assays (ELISA) in cryopreserved plasma. Namely, for hsCRP we used the Human C-Reactive Protein/CRP Quantikine^®^ ELISA Kit, R&D Systems, Inc., Minneapolis, MN, USA; for IL-6 quantification we used the Human IL-6 Standard ABTS ELISA Development Kit, PeproTech^®^, Thermo Fisher Scientific, Waltham, MA, USA; finally, for d-dimers we used the D-Dimer Human ELISA Kit, Thermo Fisher Scientific Inc., Waltham, MA, USA. Plasma samples were assayed in duplicates and according to the manufacturer’s instructions. Absorbance was measured at 450 or 405 nm, depending on the requirements of each kit. Biomarkers’ concentrations were then calculated in μg/mL based on the standard curve of each assay.

### 2.6. Assessment of Endothelial Glycocalyx Thickness

We indirectly assessed EG integrity by measuring the perfused boundary region (PBR), which is the penetrable by erythrocytes area of EG. A larger PBR indicates a deeper penetration of erythrocytes into the EG, which corresponds with a greater EG impairment [19]. Hence, PBR is inversely proportional to EG thickness.

We assessed the PBR in sublingual arterial microvessels with a diameter between 5 and 25 μm (PBR_5–25_), using a sidestream darkfield (SDF) video microscope (GlycoCheck & Microvascular Health Solutions Inc., Salt Lake City, UT, USA), enclosed in a sterile slipcover [14,20,21]. As previously described by Rick HGJ van Lanen et al. [22] “video microscope consists of a central light guide with a magnifying lens and concentric light emitting diodes. The diodes emit light at a wavelength of 530 nm, which is absorbed by (de)-oxyhemoglobin in erythrocytes. Consequently, erythrocytes appear black on a greyish background. The analysis is based on the principle of the erythrocyte-endothelial exclusion zone. The system measures the variation of the red blood cell (RBC) column’s penetration in the glycocalyx. This variation increases with a damaged or weaker glycocalyx. GlycoCheck software version 5.2 continues to collect videos until ~3000 microcirculatory vessel segments are successfully included. The complete measurements contain between 10 and 30 videos, depending on the number of vessel segments evaluated in each video. In each vessel segment, the RBC column width is measured, and vessels are automatically grouped into separate diameter classes at 1µm intervals, ranging from 5 to 25 µm in diameter.” This bedside mode of EG assessment is safe, easy to perform, and fast, as it requires only 3 min for the measurement of ~3000 sublingual microcirculatory vessel segments [14]. As higher PBR has been associated with higher prevalence of ischemic heart disease and cerebral atherosclerosis, PBR could represent a candidate bedside biomarker for CVD assessment [23].

### 2.7. Primary and Secondary Outcomes

Our primary outcome was defined as the detection of PBR changes in PLWH from baseline to 24 (±4) weeks and 48 (±4) weeks. Secondary outcomes included the comparison of baseline PBR_5–25_ between PLWH and the control group, as well as the subgroup analysis of PBR_5–25_ changes according to the sex, the treatment group, the nadir CD4+ count, the initial HIV load, the smoking status, and the illicit drug use, as well as the correlation of PBR_5–25_ changes with the kinetics of biomarkers and body weight.

### 2.8. Ethics Statement

This study was approved by the Research Ethics Committee of Attikon University Hospital (protocol number: 2887/13-12-2017) and was conducted and reported according to the “Strengthening the Reporting of Observational Studies in Epidemiology” (STROBE) Statement [24]. Participants’ data were collected and analyzed under strict anonymity in agreement with the Declaration of Helsinki.

### 2.9. Statistical Analysis

Mean ± standard deviation (SD) for normally distributed variables or median with interquartile range (IQR) for skewed data was calculated for descriptive statistics. The group comparisons were performed by the *t*-test or the non-parametric Mann–Whitney test, for two-group and ANOVA or Kruskal–Wallis for multiple groups. Qualitative variables associations were examined by the chi-square test. Quantitative variable correlations were performed by Spearman’s correlation coefficient. IBM SPSS statistical package, version 28 (IBM Software Group, New York, USA), and GraphPad Prism, version 8.0 (GraphPad Software) were used for the statistical analyses. All *p* values ≤ 0.05 were considered significant.

## 3. Results

### 3.1. Baseline Characteristics and Demographics

Demographics and characteristics are outlined in detail in Table 1. Briefly, 66 consecutive PLWH (60 (90.9%) males, 6 (9.1%) females, median age (IQR): 37 (12) years old) fulfilled the inclusion criteria and enrolled in the study. A total of 40 (60.6%) PLWH were initiated on an INSTI-based cART regimen (FTC/TAF combined with DTG or EVG/c or RAL), and 26 (39.4%) participants received a PI-based cART regimen (FTC/TAF combined with DRV/c) by the treating physician (Table 1). Baseline characteristics did not differ significantly among the two treatment groups, as shown in Table 2. Regarding follow up visits, 59 (89.4%) out of 66 PLWH attended visit 2, and 60 out of 66 (90.9%) PLWH attended visit 3.

### 3.2. Clinical Data and Routine Laboratory Results

As shown in Table 1, most of the participants (*n*=56, 84.8%) were at the second stage of HIV infection (i.e., chronic HIV infection/clinical latency, according to the classification by the Centers of Disease Control and Prevention, (CDC, [26]), with a mean (SD) nadir CD4+ T-lymphocyte count of 377.9 (244.7) cells/μL. On annual follow-up, only 4 (6.1%) PLWH had a CD4+ T-lymphocyte count of <200 cell/compared to 17 (25.8%) on visit 1. Moreover, the median (IQR) viral load before treatment initiation was 81,500 (217,695) copies/mL. Overall, 9 (13.6%) out of 66 PLWH had virological failure with detectable viral load at visit 3, potentially attributed to poor compliance with their daily medication.

The median (IQR) levels of LDL-c, HDL-c and triglycerides on enrollment visit were 108 (54) mg/dL, 40 (10) mg/dL, and 90 (68) mg/dL, respectively. Interestingly, the lipid levels increased in visit 3 by a median of 15 mg/dL (IQR 27, 95%CI: 1.2–17.9), 6.5 mg/dL (IQR 11, 95%CI: 3.9–8.7), and 26 mg/dL (IQR 58.3, 95%CI: 9.2–95.9) for LDL-c, HDL-c, and triglycerides, respectively. Of note, four participants required the initiation of lipid-lowering drugs after the 44th week of follow up; as the initiation of lipid-lowering drugs was very close to the final assessment (and thus, their effects on PBR were deemed as negligible) they were not excluded from the analysis. Finally, body weight significantly increased between the first and last visit by a median of 4 kg (IQR: 6.25, 95%CI: 3.6–6.6, *p* < 0.001).

### 3.3. Levels of Biomarkers

The median (IQR) levels of hsCRP, d-dimers and IL-6 at baseline were 2.71 μg/mL (4.09 μg/mL), 1.7 μg/mL (0.77 μg/mL), and 0.09 μg/mL (0.02 μg/mL), respectively. Additionally, we calculated the biomarker changes among 55 participants who completed both visit 1 and visit 3. We found that hsCRP levels decreased non-significantly at a median of −0.78 μg/mL (IQR: 2.47, 95%CI: (−1.51)−(+0.41)). IL-6 levels significantly dropped one year after cART initiation by a median of −0.004 μg/mL (IQR: 0.02, 95%CI: (−0.012)−(−0.003), *p* = 0.001). On the other hand, d-dimers significantly increased by 0.51 μg/mL (IQR: 0.25, 95%CI: 0.03–0.38, *p* = 0.008) one year after cART initiation.

### 3.4. Endothelial Glycocalyx Integrity Changes among All Participants after cART Initiation

Endothelial glycocalyx integrity improved significantly over the first year after cART initiation, when participants were examined as whole. Indeed, the mean PBR_5–25_ among all PLWH at baseline, 24 (±4) and 48 (±4) weeks after treatment initiation decreased significantly from 2.17 (SD: 0.29, 95%CI: 2.09–2.24) μm, to 2.04 (SD: 0.26, 95%CI: 1.97–2.11) μm (*p* = 0.019) and then to 1.93 (SD:0.30, 95%CI: 1.86–2.01) μm (*p* < 0.001), respectively (Figure 1).

### 3.5. Endothelial Glycocalyx Integrity according to Treatment Group

Neither PBR_5–25_ before treatment initiation nor PBR_5–25_ difference between the last and first visit was significantly different among the two treatment groups. Initial PBR_5–25_ was 2.13 μm (IQR 0.25, 95%CI: 2.08–2.27) among PLWH who initiated an INSTI-based regimen, while in those who started a PI-based regimen the median PBR_5–25_ was 2.18 μm (IQR 0.50, 95%CI: 2.02–2.27), (*p* = 0.16) (Figure 2). Although PBR_5–25_ decreased in both groups with a median difference of PBR_5–25_ for INSTIs and PI-based cART regimens of −0.35 (IQR: 0.54, 95% CI: (−0.45)−(−0.15)) and −0.25 (IQR: 0.67, 95% CI: (−0.34)−(+0.005)), respectively, a statistically significant drop (or otherwise a statistically significant EG improvement) was seen only in the INSTI-based treatment group (*p* < 0.001). Moreover, no significant differences were observed in PBR_5–25_ changes between the two treatment groups (*p* = 0.36).

### 3.6. Endothelial Glycocalyx Integrity according to Sex

Female participants had a higher median PBR_5–25_ compared to males in all three time points, although none of these differences were statistically significant (Table 3, Figure 3). It is evident that PBR_5–25_ decreased (i.e., EG thickness improved) at 24 (±4) and at 48 (±4) weeks after cART initiation for both sexes, but in females this reduction was not statistically significant. The difference of PBR_5–25_ between the last and first visit did not differ statistically in males and females (*p* = 0.48) (Table 3, Figure 4).

### 3.7. Endothelial Glycocalyx Integrity according to Nadir CD4+ Count and HIV Viral Load

There were no significant changes of PBR_5–25_ before and one year after treatment initiation among the different levels of nadir CD4+ T-lymphocyte counts (<200, 200–500, and >500 cells/μL, *p* = 0.761, Table 3). In contrast, initial PBR_5–25_ was significantly higher in PLWH with low levels of baseline viremia (<1000 copies/mL) compared to PLWH with viral loads of 1000–500,000 and >500,000 copies/mL (*p* = 0.017, Table 3). Additionally, there was a trend for significantly greater PBR_5–25_ difference between visits 1 and 3 in these participants, compared to the latter two groups (*p* = 0.069, Table 3, Figure 5). Finally, we compared the PBR_5–25_ at 48 (±4) weeks and the PBR_5–25_ difference between the last and the first visit among PLWH with and without virological failure to test whether incomplete compliance with treatment could affect the EG integrity; we found no statistically significant difference in PBR_5–25_ and PBR_5–25_ difference between these two groups (*p* = 0.839 and *p* = 0.411, respectively).

### 3.8. Endothelial Glycocalyx Integrity according to Smoking Status and Illicit Drug Use

We assessed PBR_5–25_ according to smoking status (smoker, non-smoker, ex-smoker) on enrollment day. Smokers had the highest PBR_5–25_ (median: 2.17 μm, IQR: 0.22, 95%CI: 2.10–2.26), followed by non-smokers and ex-smokers (Table 3). The greatest PBR_5–25_ change was detected among smokers (median: −0.37 μm, IQR: 0.53, 95%CI: (−0.44)−(−0.17)); this statistically significant reduction in PBR_5–25_ (*p* <0.001), shows a definite improvement of EG over the first year of cART initiation in this group. Non-smokers and ex-smokers demonstrated a trend of PBR_5–25_ reduction over the first year, but it was not statistically significant (Table 3). However, the population of smokers was much larger than non-smokers and ex-smokers in our cohort (35 vs. 18 vs. 6, respectively, among those with available data for both first and last visits). Although we based this subgroup analysis on the baseline smoking status, it should be mentioned that smoking status remained unchanged for all participants during the study period.

In addition to smoking, we analyzed PBR_5–25_ according to the use of illicit/recreational drugs (including chemsex). Of the 66 participants, 22 (33.3%) reported using illicit/recreational drugs. Initial PBR_5–25_ thickness was comparable between PLWH who used and did not use illicit/recreational drugs, and the median progress of PBR_5–25_ did not differ between the two groups (Table 3).

### 3.9. Correlation between Reduction in PBR_5–25_ over the First Year and Other Variables

In this cohort, there was no correlation between the magnitude of PBR_5–25_ reduction over the first year of cART introduction and the baseline levels of hsCRP, d-dimers, and IL-6 (Spearman’s rho *p* values: 0.579, 0.920 and 0.221, respectively). Similarly, there was no correlation between PBR_5–25_ change in visits 1 and 3 and the kinetics of these biomarkers (Spearman’s rho *p* values: 0.254, 0.248 and 0.739, respectively).

As with biomarker levels, no correlation was found between the reduction in PBR_5–25_ thickness during the first year of cART and body weight increase (Spearman’s rho *p* = 0.962), nadir CD4+ count (Spearman’s rho *p* = 0.426), and baseline HIV viral load (Spearman’s rho *p* = 0.096).

## 4. Discussion

In this study, we demonstrated that PBR_5–25_ is significantly reduced (i.e., EG integrity is significantly improved) at both 6 and 12 months after cART initiation in treatment-naïve PLWH. Interestingly, smokers had the greatest benefit in terms of PBR_5–25_ reduction, compared to non-smokers and ex-smokers. Moreover, PLWH on an INSTI-based regimen experienced a statistically significant reduction in PBR_5–25_ at 48(±4) weeks, although no significant differences were observed between the two treatment groups in terms of initial PBR_5–25_ and PBR_5–25_ differences over the first year. However, subgroup analysis for sex, illicit drug use, initial HIV viral loads, and the level of nadir CD4+ count did not show statistically significant change of PBR_5–25_ among different groups. Finally, in this cohort we did not find any correlation between hsCRP, d-dimers and IL-6 kinetics and PBR_5–25_ evolution within the first year of cART initiation.

To our knowledge, this is the first study reporting on the trajectory of EG thickness in an HIV population after cART initiation, using side stream darkfield imaging. In 2017, Meneses et al. studied the association between the EG integrity and renal dysfunction in PLWH without overt renal disease by measuring the levels of syndecan-1, an indirect marker of glycocalyx damage [27]. Syndecan-1 was higher in PLWH compared with healthy controls, and especially in PLWH under cART (i.e., in those receiving tenofovir, followed by zidovudine, then by non-treated individuals), and was independently associated with higher serum creatinine and reduced glomerular filtration rate after adjustments for variables related with HIV infection in a multivariate analysis [27]. The authors concluded that PLWH on cART present endothelial glycocalyx damage which is associated with clinical markers of kidney dysfunction [27]. A more recent study by Cavalcante et al. reported similar results, with syndecan-1 showing a significant correlation with serum creatinine and glomerular filtration rate in PLWH under long-term cART over a 5-year follow up period [28].In contrast to these studies, our cohort showed a significant improvement of EG thickness after cART initiation over a longitudinal follow-up of 12 months, supporting the hypothesis that viral suppression and immunological reconstitution improves EG, at least during the first year since cART introduction. It should be highlighted that all participants were given a tenofovir alafenamide-based cART backbone regimen (FTC/TAF) to minimize potential confounder effects. Additionally, in both studies, treated individuals were on a cART regimen for at least 14 months. Whether the observed EG improvement in our study will be sustained beyond the first year of treatment, or other factors, such as the cumulative exposure to cART and/or the co-existing persistent immune activation, will contribute to a relapse of endothelial dysfunction and a new reduction in EG thickness, remains to be answered in future studies.

In our study we found that EG improves significantly within the first year of cART initiation in PLWH. However, it is very important to clarify whether this reduction represents a return to a “normal” EG integrity (or to PBR_5–25_ values that are comparable to these of general population) or EG derangement persists even after the initiation of cART initiation and the control of HIV viremia. Previous studies conducted in our center, have shown that PBR_5–25_ values in healthy adult controls (slightly older than our participants) ranges between 1.77 and 1.78 μm [29,30]. Similarly, a small observational study of healthy young adults measuring PBR_5–25_ before and after exercise, reported a baseline PBR_5–25_ of 1.86 μm [31]. Compared to the results of PBR_5–25_ at 48 ± 4 weeks (1.93 μm) in our cohort, these values show that EG integrity is not completely restored among PLWH after one year of cART. Hence, cART does improve the microvascular effects of HIV infection but only partially (at least within the first year of treatment), and other factors of endothelial damage (such as chronic immune activation) should also be taken under consideration in HIV infection.

Interestingly, our study showed that females had higher PBR_5–25_ measurements compared to men in all three time points; however, this difference did not reach statistical significance. Nevertheless, we cannot draw definite conclusions on the differences of EG integrity between the two sexes from our cohort, as the female sex was underrepresented. Evidence from both preclinical and clinical studies indicates that females may indeed have a higher degree of EG dysfunction compared to males. In a study comparing sublingual microvascular perfusion and glycocalyx barrier properties in people with coronary artery disease (CAD) and healthy controls showed that CAD was significantly associated with impaired sublingual microvascular glycocalyx barrier function in women but not in men [32]. Likewise, another study assessing the association of syndecan-4 levels with myocardial infarction, ischemic stroke, and all-cause mortality, found that syndecan-4 was associated with myocardial infarction only in women, suggesting a potential link between EG shedding and coronary artery disease in women [33]. Additionally, a murine model showed the presence of a gender difference in systemic glycocalyx volume, with female mice presenting a 50% reduction [34]. It is evident that sex affects EG both structurally and functionally, but data are still limited, and more studies are needed to evaluate the influence of sex on EG.

In this study we found that smokers had the greatest degree of EG recovery one year after cART initiation. Smoking represents one of the most important modifiable cardiovascular risk factors [35]. Tobacco exposure has been associated with as many as 36 fatal and non-fatal subtypes of CVD [36]. Among PLWH, smoking augments the possibility of developing HIV-related and unrelated comorbidities [37]. PLWH are more likely to develop the habit of smoking, to smoke more cigarettes per day, and are less likely to quit tobacco use than the general population [38]. Yet, there are no studies in this population evaluating the impact of tobacco exposure on microvasculature. Expectedly, in our cohort, smokers had initially the most affected EG compared to non-smokers and former smokers. However, smokers showed the greatest PBR_5–25_ reduction at 48 (±4) weeks after cART initiation among all three subgroups. This outcome may be possibly explained by the fact that smokers’ microvasculature was initially “more affected” compared to the other subgroups, as endothelium was simultaneously exposed to two important insults (HIV and tobacco); by detracting a significant factor like active HIV replication and viremia in those exposed to tobacco may have resulted in a significant PBR reduction, as demonstrated in this study. Nonetheless, it should be highlighted that the population of all three subgroups was not equally distributed, with the vast majority being smokers. Hence, no definite conclusions can be drawn regarding the effects of smoking on EG integrity among PLWH after cART initiation.

In addition to smoking, HIV-related factors, such as cART exposure and HIV-induced immune activation, have also been associated with increased CVD risk in PLWH [5]. We examined the effects of two different cART groups on EG integrity; one group received a cobicistat-boosted PI and the other received three different INSTIs. In both groups, PI and INSTIs were combined with the same backbone of FTC/TAF. While PIs have a well-documented class-related toxicity on the cardiovascular system, INSTIs, a relatively new class of antiretrovirals, have a better cardiometabolic profile, despite their effect on bodyweight [5]. Based on these data, we hypothesized that EG would be more “severely affected” in the PI group, as well as that EG improvement would be greater in the INSTI group. However, our hypothesis was not confirmed by this study, as there was no statistically significant difference of PBR_5–25_ between the two groups. Although this can be explained by the small number of participants per cART group, one should keep in mind that the period of exposure to each cART regimen was not long enough to bring out any potential differences among the two groups. Hence, a longer follow up is warranted.

Based on increasing evidence showing that HIV-related injury of endothelial cells is associated with immune activation driven by HIV-infected cells [39,40,41], we hypothesized that EG thickness will be inversely proportional to the levels of hsCRP, d-dimers and IL-6. Even though this study failed to show any correlation between PBR_5–25_ and the kinetics of the biomarker levels, this line of inquiry should be further explored in larger studies.EG plays a pivotal role in protecting endothelium from various insults [42,43]. Several risk factors for atherosclerosis have been linked to EG derangement and shedding [44], while endothelial dysfunction has been associated with the development of CVD [45]. In this context, EG can represent both a biomarker of CVD [46], as well as a candidate therapeutic target for CVD [44]. In HIV infection, endothelium represents a key target of viral proteins and inflammatory molecules produced by the HIV-infected cells [47]; hence, HIV is associated with endothelial disruption and dysfunction, which, in turn, can lead to CVD development; however, the role of EG structure and function in HIV infection has been hugely neglected.

On the contrary, its central role in the pathogenesis of several other chronic or acute conditions, including other infections, has been documented in scientific literature. In a murine model studying the pathogenesis of malaria, researchers directly visualized in light microscopy a significant loss of EG in brain tissue by mice with cerebral malaria [48]. Although this phenomenon was absent in the brain tissue of mice with uncomplicated malaria, the researchers detected high plasma levels of EG components which may depict the degradation and shedding of EG in other organs [48,49]. Moreover, an ex vivo study on human umbilical vein endothelial cells demonstrated that H1N1 influenza virus induces endothelial dysfunction by causing EG degradation in a dose-dependent fashion [50]. This result underlines that not only the presence of the virus is important for EG destruction, but also that the degree of EG impairment is proportional to the circulating viral loads. In contrast to this study, we found that PLWH with the lowest levels of HIV viremia experienced the greatest PBR improvement over time after treatment initiation, showing that, within the context of HIV infection, the insult to EG may not be directly mediated from the virus itself, but from the associated activation of the immune system. Although this finding was statistically significant, it should be highlighted that the number of PLWH with the lowest (<1000 copies/mL) and highest (>500,000 copies/mL) initial viral loads was very small, and therefore a cautious interpretation of this result is warranted.

Additionally, a recent experimental study investigating the pathogenesis of coronavirus disease 2019 (COVID-19) by Targosz-Korecka et al., demonstrated that EG has a significant role in regulating the binding of Severe Acute Respiratory Syndrome coronavirus 2 (SARS-CoV-2) spike protein to the endothelial surface [51]. Finally, it has been shown that EG cleavage and shedding caused by several enzymes and mechanisms that are driven by the pro-inflammatory milieu plays an important role in the pathogenesis of sepsis [52]; EG components can be both surrogate markers of endothelial barrier damage and mechanistic contributors of multi-organ damage in sepsis [52].

Based on these data, EG derangement can either be the cause and/or the result of endothelial damage in these conditions. Hence, evaluating the magnitude of EG destruction (through measuring the PBR) is both clinically and biologically meaningful, as depicts the damage on a microvascular level. Although the role of glycocalyx in endothelial protection has long been acknowledged, there are very limited data demonstrating that the numerical quantification of EG impairment could correlate with outcomes, such as the disease severity, morbidity, mortality, etc., in non-HIV population. Hence, there is currently no numerical “cut off” value for PBR that could evoke certain clinical decisions. Further studies are warranted to establish whether there is a role of such “cut-offs” in clinical practice.

Various approaches for glycocalyx regeneration have been pursued in the context of reversing endothelium damage leading to atherosclerotic plaque formation. Most commonly, atherosclerotic plaque stabilizers (e.g., sulodexide and rosuvastatin), anti-diabetic drugs (metformin), anti-inflammatory therapies, anticoagulants, and dietary supplements have been investigated, both in animal and human studies, with some early but promising results [44,53]. Glycocalyx repeated measurements could also be employed to assess the response to existing treatments for CVD. However, there still are barriers in the widespread use of glycocalyx measurement, that are related to assay precision, clinical interpretation of findings, and lack of homogeneity between different assessment methods [54]. Hence, there is still a long way (in terms of research) before EG integrity evaluation could be established as a routine test for CVD assessment in PLWH.

Our study has several strengths. First and foremost, this is the first study assessing the integrity of EG in PLWH before and one year after cART initiation. Reporting the changes of EG in this population, not only sheds light into a possible pathophysiologic mechanism of CVD development among HIV positive people, but also provides data for a possible surrogate biomarker of CVD which can be easily and efficiently assessed. Moreover, the assessment of EG thickness was performed by side stream darkfield imaging that measures the dimensions of PBR in sublingual microvasculature. This method is simple, non-invasive, and fast, with a good intra- and inter-observer reproducibility and with practically no adverse effects or risks for the patient [55,56]. Compared to the measurement of the plasma levels of EG components, which indirectly assesses the degree of glycocalyx cleavage from several tissues, this technique is certainly more specific to endothelial glycocalyx; another benefit is that it quantifies the actual dimensions of PBR, while measurements of EG components plasma levels can only qualitatively be associated with the degree of EG destruction. However, side stream darkfield imaging can be used only in a handful of body areas to directly visualize and measure PBR dimensions. In addition to this disadvantage, the cost of the device and the training that is required to perform the measurements of PBR should also be kept in mind.

Another advantage of our study is the carefully selected inclusion and exclusion criteria that intended to eliminate as much as possible several confounders that may affect EG integrity, such as hyperglycemia and uncontrolled hypertension. Through this participant selection process, we aimed to examine the (net) effects of HIV infection and cART initiation on PBR. For the same reason we selected treatment-naïve PLWH who were put on two groups of cART regimens with the same backbone of FTC/TAF.

This study also has some limitations. First, the number of participants is relatively small; whilst results showed a statistically significant reduction in PBR when the study population was examined as a whole, most subgroup analyses failed to reach statistical significance due to the small number of participants. Furthermore, females were underrepresented in our study; as we recruited consecutive PLWH, it was inevitable that the proportion of females in this study followed the actual proportion of female participants in our unit (90% males and 10% females). Third, the PLWH were not equally distributed between the two treatment regimens, albeit baseline characteristics of both groups did not significantly differ. Additionally, the follow-up period in our study was short; although we identified significant changes of EG within the first year of cART introduction, the association of these changes with the development or regression of CVD, requires a much longer follow-up period. Finally, when this study protocol was designed, both INSTIs and PIs were proposed as first line agents. However, since 2020, PIs are no longer recommended as first line agents, but they still represent an alternative that can be used in specific cases [57,58].

## 5. Conclusions

EG plays a cardinal role in protecting endothelial cells’ function and prosperity. Hence, insults affecting the structure and the function of EG inevitably affect endothelium, a phenomenon that has been associated with CVD development in the long term [59]. In this study, we showed for the first time that EG thickness progressively improves one year after cART introduction in treatment-naïve PLWH. This outcome not only depicts a potential pathogenetic mechanism of CVD among PLWH, but also reveals a candidate biomarker and a possible therapeutic target for (micro)vascular disease in the specific population. However, large longitudinal studies are certainly needed to further explore and establish the multifaceted role of EG in HIV infection.

## Figures and Tables

**Figure 1 viruses-15-01505-f001:**
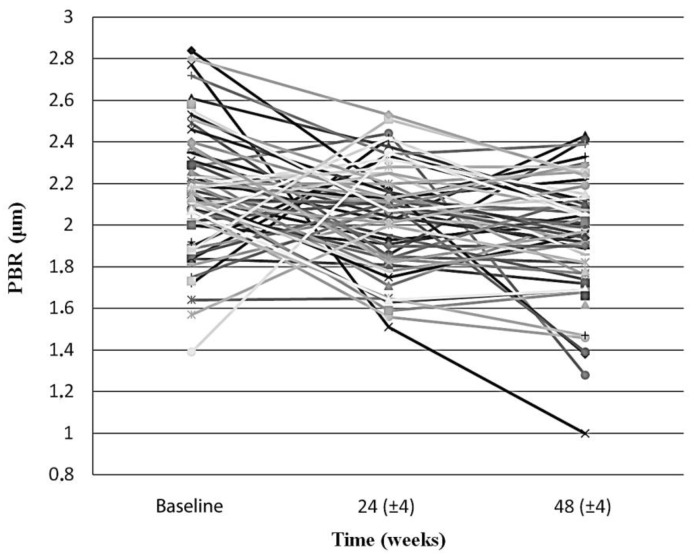
The slope chart shows the PBR_5–25_ (in μm) of each individual participant at baseline, at 24 (±4) weeks and at 48 (±4) weeks after initiation of antiretroviral treatment. PBR_5–25_ decreased in the majority of the participants in both timepoints, compared to baseline (66% and 70%, for 24 (±4) and 48 (±4) weeks, respectively). Sex, age, antiretroviral treatment, body weight, and smoking status did not differ significantly between the two groups (i.e., those whom PBR_5–25_ increased and those whom PBR_5–25_ decreased compared to the baseline) in both timepoints. PBR_5–25_: Permeability boundary region in sublingual microvessels with dimeter between 5 and 25 μm.

**Figure 2 viruses-15-01505-f002:**
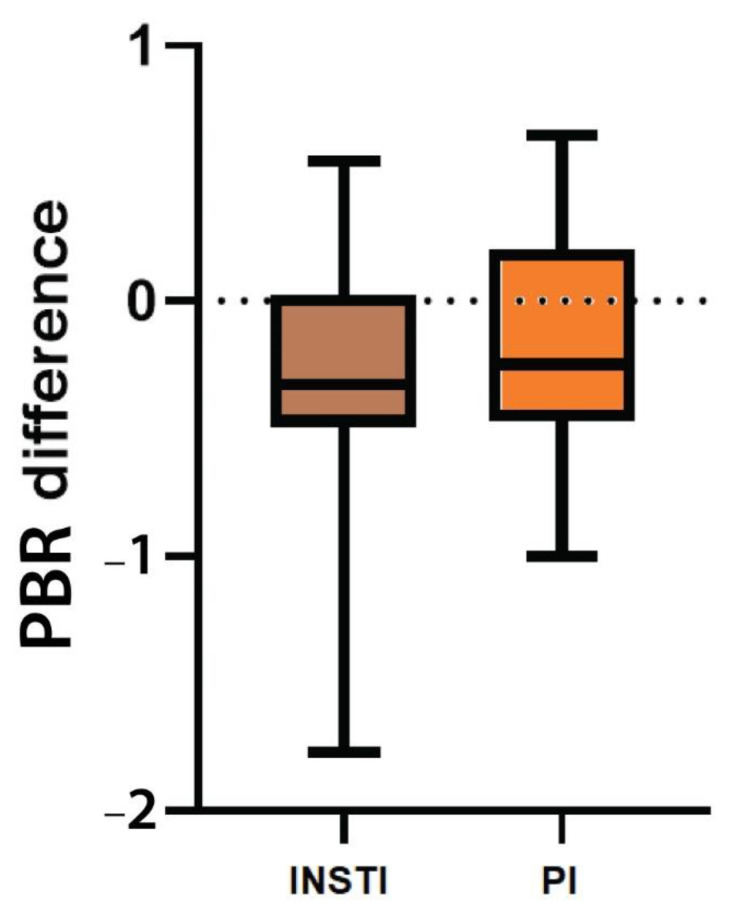
Box plot shows the difference of PBR_5–25_ (in μm) between the last (48 ± 4 weeks after treatment initiation) and the baseline visit among the two different treatment groups. INSTI: Integrase strand transfer inhibitors, PBR_5–25_: Permeability boundary region in sublingual microvessels with dimeter between 5 and 25 μm, PI: Protease inhibitor.

**Figure 3 viruses-15-01505-f003:**
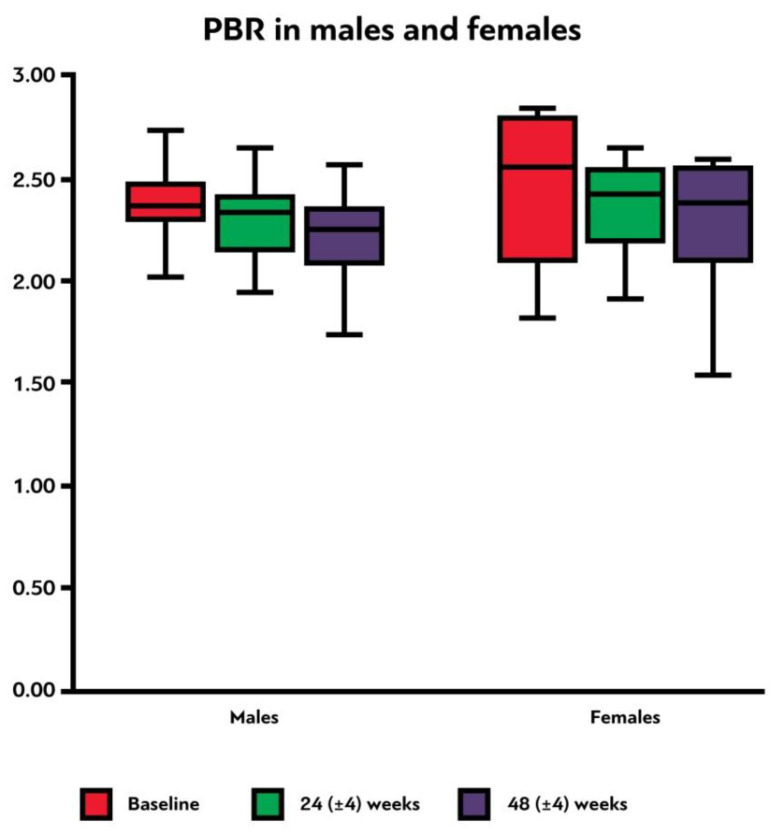
Box plot shows the PBR_5–25_ thickness (in μm) at baseline, 24 (±4) weeks and 48 (±4) weeks after initiation of antiretroviral treatment in male and female participants. PBR_5–25_: permeable boundary region of sublingual microvessels with diameter between 5 and 25 μm.

**Figure 4 viruses-15-01505-f004:**
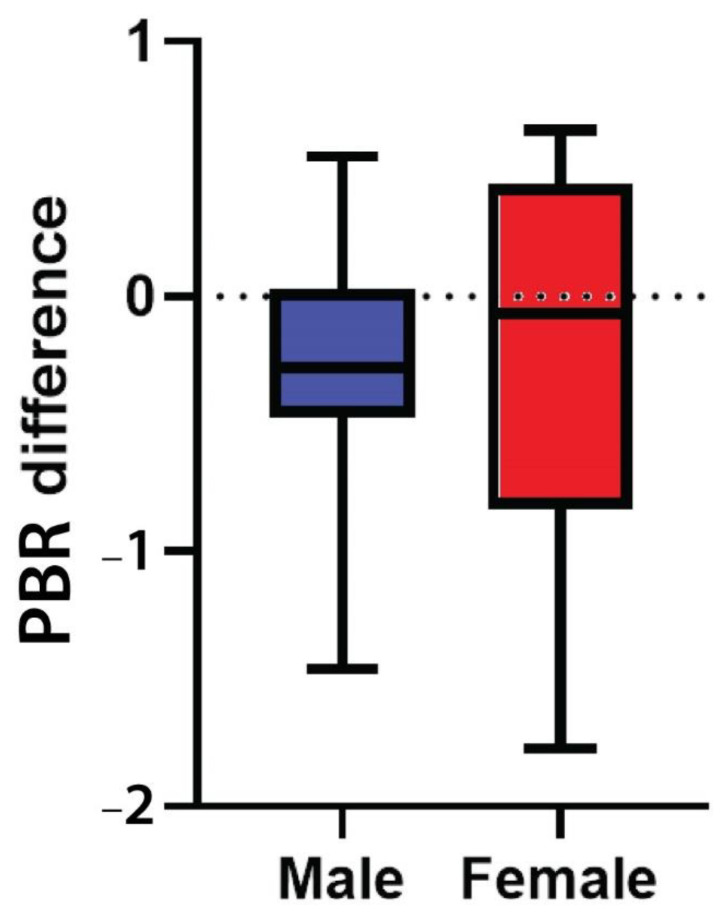
Box plot shows the difference of PBR_5–25_ thickness (in μm) between the last (48 ± 4 weeks after treatment initiation) and the baseline visit in male and female participants. PBR_5–25_: permeable boundary region of sublingual microvessels with diameter between 5 and 25 μm.

**Figure 5 viruses-15-01505-f005:**
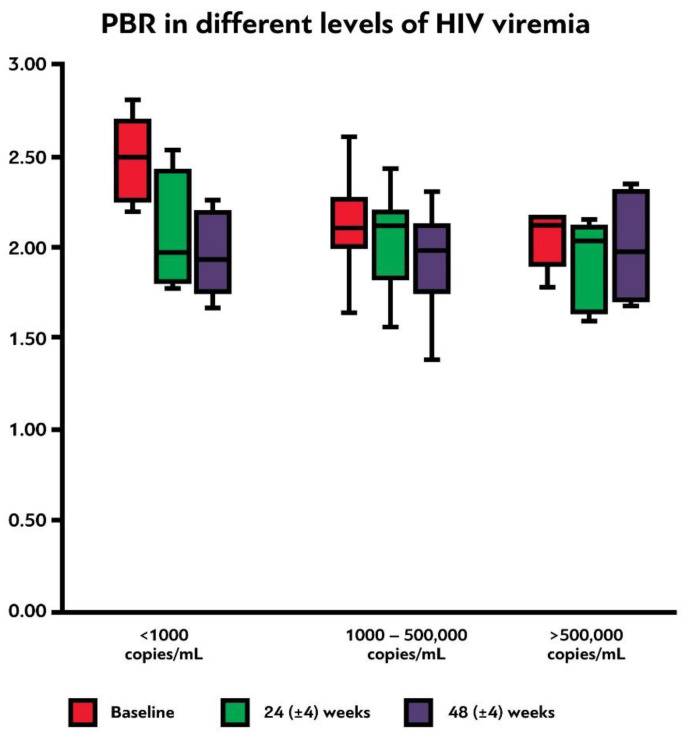
Box plot shows the PBR_5–25_ (in μm) at baseline, 24 (±4) weeks and 48 (±4) weeks after initiation of antiretroviral treatment among participants with different initial levels of HIV viremia. PBR_5–25_: permeable boundary region of sublingual microvessels with diameter between 5 and 25 μm.

**Table 1 viruses-15-01505-t001:** Demographics and baseline characteristics.

GENERAL CHARACTERISTICS	
Sex (*n* (%))	
Male	60 (90.9)
Female	6 (9.1)
Age (years) (median (IQR))	37 (12.0)
Antiretroviral regimen (*n* (%))	
INSTIs	40 (60.6)
*DTG*	*24 (36.4)*
*RAL*	*7 (10.6)*
*EVG/c*	*9 (13.6)*
PI	26 (39.4)
*DRV/c*	*26 (39.4)*
HIV stage (*n* (%)) ^a^	
Stage 1	7 (10.6)
Stage 2	42 (63.6)
Stage 3	17 (25.8)
Nadir CD4+ T-lymphocytes (cells/μL) (mean (SD))	377.9 (244.7)
Nadir CD4+ T-lymphocytes (cells/μL) (*n* (%))	
<200	17 (25.8)
200–500	29 (43.9)
>500	20 (30.3)
Peak viral load (copies/mL) (n (%)) ^b^ [Participants with available data: N = 53]	
<1000	5 (9.4)
1000–500,000	41 (77.4)
>500,000	7 (13.2)
Race (*n* (%))	
Caucasian	61 (92.4)
African	3 (4.6)
Other	2 (3.0)
Mode of HIV transmission (*n* (%))	
MSM	41 (62.1)
Heterosexual	21 (31.8)
IVDU	3 (4.5)
Other/Unknown	1 (1.5)
Educational stage (*n* (%))	
Unknown	3 (4.5)
Primary school	7 (10.6)
Secondary school	13 (19.7)
Higher education/University	43 (65.2)
BMI (kg/m²) (median (IQR)) ^c^	24.2 (3.86)
**TRADITIONAL CVD RISK FACTORS**	
Smoking status (*n* (%)) ^c^	
Yes	40 (60.6)
No	19 (28.8)
Ex smoker	7 (10.6)
Family history of premature CVD (*n* (%))	
Yes	10 (15.2)
No	56 (84.8)
Hypertension (*n* (%)) ^c^	
Yes	2 (3.0)
No	64 (97.0)
Diabetes mellitus (*n* (%)) ^c^	
Yes	0 (0.0)
No	66 (100.0)
Dyslipidemia (*n* (%)) ^c^	
Yes	3 (4.5)
No	63 (95.5)

BMI: body mass index; CVD: cardiovascular disease; DRV/c: cobicistat boosted darunavir; DTG: dolutegravir; EVG/c: cobicistat boosted elvitegravir; HIV: human immunodeficiency virus; INSTIs: integrase strand transfer inhibitors; IQR: interquartile range; IVDU: intravenous drug users; MSM: men having sex with men; PIs: protease inhibitors; RAL: raltegravir; SD: standard deviation, VL: viral load. a. Based on the classification by the Centers for Disease Control and Prevention (CDC) [25]. b. Percentages have been calculated in the population with available data. c. Data from visit 1 (enrollment visit). Italics denote specific types of antiretrovirals used in combination with tenofovir alafenamide and emtricitabine.

**Table 2 viruses-15-01505-t002:** Participants’ characteristics per treatment group.

	INSTIs *N =* 40	PI *N =* 26	*p* Value
Age (years) (median (IQR))	37.0 (13.0)	37.5 (13.0)	0.595
Sex (*n* (%))			0.202
Male	38 (95.0)	22 (84.6)	
Female	2 (5.0)	4 (15.4)	
Smoking status (*n* (%)) *			0.521
Yes	26 (65.0)	14 (53.8)	
No	11 (27.5)	8 (30.8)	
Ex smoker	3 (7.5)	4 (15.4)	
LDL-c (mg/dL) (mean (SD)) *	119.6 (40)	105.4 (38.5)	0.158
HDL-c (mg/dL) (median (IQR)) *	41.0 (14)	38.0 (7)	0.205
Triglycerides (mg/dL) (median (IQR)) *	91.0 (64)	89.0 (70)	0.634
Body weight (kg) (mean (SD)) *	74.2 (10.4)	77.2 (16.7)	0.417
BMI (kg/m²) (mean (SD)) *	24.0 (2.3)	25.0 (5.5)	0.397
Nadir CD4+ T-lymphocytes (cells/μL) (mean (SD)) *	412.5 (240.9)	324.8 (245.6)	0.157
Viral load before cART initiation (copies/mL) (median (IQR)) *	41,100 (228,020)	111,000 (192,250)	0.467
Lost to follow up (*n* (%))			-
24 (±4) weeks	4 (10.0)	3 (11.5)	
48 (±4) weeks	3 (7.5)	3 (11.5)	

BMI: body mass index; cART: combination antiretroviral therapy; HDL-c: high density lipoprotein cholesterol; INSTIs: integrase strand transfer inhibitors; IQR: interquartile range; LDL-c: low density lipoprotein; PI: protease inhibitor; SD: standard deviation. * Data form visit 1 (enrollment visit).

**Table 3 viruses-15-01505-t003:** Subgroup analyses of median PBR_5–25_ according to sex, nadir CD4+ count, initial viral load, smoking, and illicit drug use.

Subgroups	Median PBR_5–25_ (IQR) in μm at Baseline (95% CI)	Median PBR_5–25_ (IQR) in μm at 24 (±4) Weeks (95% CI)	Median PBR_5–25_ (IQR) in μm at 48(±4) Weeks (95% CI)	Median PBR_5–25_ (IQR) Difference in μm (95% CI) *
Sex				
Male	2.13 (0.28) (2.09–2.22)	2.08 (0.35) (1.96–2.10)	1.96 (0.36) (1.85–1.99)	−0.28 (0.51) [(−0.35)–(−0.15)]
Female	2.38 (0.98) (1.68–2.82)	2.21 (0.50) (1.76–2.50)	2.15 (0.64) (1.46–2.58)	−0.07 (1.27) [(−1.15)–(+0.68)]
Nadir CD4+ count (cells/μL)				
<200	2.21 (0.40) (2.02–2.40)	2.13 (0.30) (1.96–2.26)	2.11 (0.40) (1.86–2.22)	−0.15 (0.68) [(−0.39)–(+0.04)]
200–500	2.07 (0.20) (1.96–2.21)	2.03 (0.50) (1.91–2.18)	1.97 (0.33) (1.85–2.05)	−0.28 (0.57) [(−0.33)–(−0.01)]
>500	2.24 (0.68) (2.05–2.45)	2.11 (0.36) (1.87–2.19)	1.97 (0.34) (1.71–2.08)	−0.33 (0.55) [(−0.67)–(−0.11)]
Initial viral load (copies/mL)				
<1000	2.52 (0.37) (2.26–2.71)	1.96 (0.62) (1.52–2.59)	1.92 (0.45) (1.66–2.24)	−0.56 (0.30) [(−0.71)–(−0.31)]
1000–500,000	2.11 (0.37) (2.01–2.21)	2.12 (0.45) (1.98–2.15)	2.01 (0.33) (1.87–2.05)	−0.18 (0.52) [(−0.31)–(−0.06)]
>500,000	2.12 (0.27) (1.85–2.34)	2.06 (0.49) (1.57–2.23)	1.94 (0.62) (1.66–2.29)	−0.12 (0.93) [(−0.67)–(+0.45)]
Smoking Status				
Smokers	2.17 (0.22) (2.10–2.26)	2.03 (0.42) (1.87–2.08)	1.89 (0.39) (1.77–1.99)	−0.37 (0.53) [(−0.44)–(−0.17)]
Non-smokers	2.13 (0.53) (1.99–2.36)	2.10 (0.35) (1.98–2.26)	2.05 (0.19) (1.91–2.18)	−0.27 (0.73) [(−0.45)–(+0.03)]
Ex smokers	2.02 (0.32) (1.82–2.25)	2.33 (0.51) (1.87–2.59)	2.08 (0.29) (1.93–2.31)	−0.04 (0.39) [(−0.23)–(+0.26)]
Illicit drug use				
Yes	2.12 (0.21) (1.99–2.21)	1.96 (0.51) (1.82–2.17)	1.89 (0.45) (1.71–2.02)	−0.33 (0.54) [(−0.40)–(−0.11)]
No	2.22 (0.36) (2.10–2.29)	2.12 (0.41) (1.99–2.18)	2.01 (0.28) (1.91–2.09)	−0.12 (0.57) [(−0.40)–(−0.05)]

CI: confidence interval; IQR: interquartile range; PBR_5–25_: permeable boundary region of sublingual microvessels with diameter between 5 and 25 μm. * Difference refers to the difference of PBR_5–25_ at 48(±4) weeks (visit 3) and at baseline (visit 1).

## Data Availability

Data can be provided upon reasonable request from the corresponding author.

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
