# Peer review of "Endothelial Glycocalyx Integrity in Treatment-Naïve People Living with HIV before and One Year after Antiretroviral Treatment Initiation"

_viruses, 2023, doi:10.3390/v15071505_

Round 1
Reviewer 1 Report
Endothelial glycocalyx (EG) derangement associates with cardiovascular disease. EG can be estimated by measuring perfused boundary region (PBR) in the sublingual microvessels. PBR inversely correlates with EG derangement. Investigators prospectively followed 60 antiretroviral naïve people with HIV (PWH) at baseline and 24 and 48 weeks after antiretroviral treatment (ART) initiation. They find that PBR decreases with ART initiation suggesting that decreasing virus levels improves EG derangement. Investigators also do sub group analysis and show some differences among genders and those on different ART. This study has potentially interesting finding although the results and conclusions need to be more precise and less generalized One of the primary issues is that the clinical/biological significance of the observed PBR differences is not well defined. The PWH were primarily male and mostly placed on integrase inhibitor therapy. In many instances, the study was not designed and did not have statistical power for a number of the results and conclusions. In addition, it may be ideal to conduct a multi-variate analysis to understand some of the factors that EG derangement in PWH after antiretroviral therapy.
1) Please describe PBR measurement/methodology in greater detail.
2) Please specify if any individual was on a cholesterol medication. This presumably impacts EG or PBR measurements.
3) Figure 1 and figure 2 show that PBR did not go down in all instances. Can the investigators provide greater detail among the PWH where PBR did not go down especially from baseline to 24 week measurement.
4) There is a decrease in PBR. Is this biologically or clinically meaningful? What level of decrease is associated with a marker of clinical improvement?
5) I would remove the sections and discussion about PBR measurement differences based on integrase versus protease inhibitor differences and men versus women. The study was never designed to examine these differences. It does not have statistical power to make any statements regarding these confounders.
6) It is surprising that PBR is lower among those with higher as compared to lower virus level at baseline. If virus is driving EG derangement, I would have expected the opposite. How do investigators explain this?
7) The study likely does not statistical power to detect differences among smokers versus non-smokiers. Was the smoking status static over the 48 weeks? In other words, those classified as smokers continued to smoke over the 48 weeks and similarily for the non-smokers and ex-smokers?
Appropriate
Author Response
Dear Editor and Reviewers,
Please find attached the revised manuscript entitled “Endothelial glycocalyx integrity in treatment-naïve people living with HIV before and one year after antiretroviral treatment initiation.” (MS ID#: viruses-2411319). On behalf of all co-authors, we greatly appreciate both Reviewers’ comments and wish to thank them both for their time and valuable input into our manuscript. In this revised submission we have addressed all comments and suggestions made by the Reviewers. Our point-by-point responses to each comment are provided below. The page and line numbers refer to the clean version of the manuscript.
REVIEWER #1
Comment 1
Endothelial glycocalyx (EG) derangement associates with cardiovascular disease. EG can be estimated by measuring perfused boundary region (PBR) in the sublingual microvessels. PBR inversely correlates with EG derangement. Investigators prospectively followed 60 antiretroviral naïve people with HIV (PWH) at baseline and 24 and 48 weeks after antiretroviral treatment (ART) initiation. They find that PBR decreases with ART initiation suggesting that decreasing virus levels improves EG derangement. Investigators also do subgroup analysis and show some differences among genders and those on different ART. This study has potentially interesting finding although the results and conclusions need to be more precise and less generalized One of the primary issues is that the clinical/biological significance of the observed PBR differences is not well defined. The PWH were primarily male and mostly placed on integrase inhibitor therapy. In many instances, the study was not designed and did not have statistical power for a number of the results and conclusions. In addition, it may be ideal to conduct a multi-variate analysis to understand some of the factors that EG derangement in PWH after antiretroviral therapy.
Authors’ response
We thank the Reviewer for these general comments. We agree that the study was powered in order to detect PBR thickness trajectory in the whole cohort, which was our primary outcome, and not for subgroup differences (i.e., our secondary outcomes). Nevertheless, we believe that these secondary outcomes have an exploratory character and could provide a basis for further investigation. We address in detail this and the other points by the Reviewer 1 under individual comments below. Additionally, we added a new paragraph in our methods detailing our primary and secondary outcomes that will help the readers to better evaluate our results.
Page: 4, Lines: 168-174: “2.7. Primary and secondary outcomes
Our primary outcome was defined as the detection of PBR changes in PLWH from baseline to 24 (±4) weeks and 48 (±4) weeks. Secondary outcomes included the subgroup analysis of PBR changes according to the sex, the treatment group, the nadir CD4+ count, the initial HIV load, the smoking status and the illicit drug use, as well as the correlation of PBR changes with the kinetics of biomarkers and body weight.”
Comment 2
Please describe PBR measurement/methodology in greater detail.
Authors’ response
We are grateful for pointing this out. In response to this comment, we have modified the respective section of our “Materials and Methods” as follows:
Page: 4, Lines: 148-167: “We assessed the PBR in sublingual arterial microvessels with diameter between 5 and 25 μm (PBR5-25), using a sidestream darkfield (SDF) video microscope (GlycoCheck & Microvascular Health Solutions Inc., Salt Lake City, UT, USA), enclosed in a sterile slipcover [14,20,21]. As previously described by Rick HGJ van Lanen et al. [22]: “video microscope consists of a central light guide with a magnifying lens and concentric light emitting diodes. The diodes emit light at a wavelength of 530 nm, which is absorbed by (de)-oxyhemoglobin in erythrocytes. Consequently, erythrocytes appear black on a greyish background. The analysis is based on the principle of the erythrocyte-endothelial exclusion zone. The system measures the variation of the red blood cell (RBC) column’s penetration in the glycocalyx. This variation increases with a damaged or weaker glycocalyx. GlycoCheck software continues to collect videos until ∼3000 microcirculatory vessel segments are successfully included. The complete measurements contain between 10 and 30 videos, depending on the number of vessel segments evaluated in each video. In each vessel segment, the RBC column width is measured, and vessels are automatically grouped into separate diameter classes at 1 µm intervals, ranging from 5 to 25 µm in diameter.” This bedside mode of EG assessment is safe, easy to perform and fast, as it requires only 3 minutes for the measurement of ∼3000 sublingual microcirculatory vessel segments [14]. As higher PBR has been associated with higher prevalence of ischemic heart disease and cerebral atherosclerosis, PBR could represent a candidate bedside biomarker for CVD assessment [23].”
Comment 3
Please specify if any individual was on a cholesterol medication. This presumably impacts EG or PBR measurements.
Authors’ response
We greatly appreciate Reviewer 1 for pointing this out. Indeed, there is some evidence that statins may (at least partially) induce recovery of endothelial glycocalyx in patients with familial hypercholesterolemia [Meuwese, Marijn C et al. “Partial recovery of the endothelial glycocalyx upon rosuvastatin therapy in patients with heterozygous familial hypercholesterolemia.” Journal of lipid research vol. 50,1 (2009): 148-53.]. Although this assumption made by Reviewer 1 is (admittedly) very logical, the data on the effects of lipid-lowering drugs on endothelial glycocalyx in general population are extremely limited. Moreover, in our cohort only 3 participants initiated a statin and 1 initiated evolocumab, after the 44th week of follow up. Hence, it is impossible that lipid-lowering drugs could have affected our results. For all these reasons, we decided not to proceed with a subgroup analysis on PLWH who initiated a lipid-lowering drug or exclude them from the analysis. With regards to this comment, we added the following in our results:
Page: 7, Lines: 220-223: “Of note, 4 participants required the initiation of lipid-lowering drugs after the 44th week of follow up; as the initiation of lipid-lowering drugs was very close to the final assessment (and thus, their effects on PBR were deemed as negligible) they were not excluded from the analysis.”
Comment 4
Figure 1 and figure 2 show that PBR did not go down in all instances. Can the investigators provide greater detail among the PWH where PBR did not go down especially from baseline to 24 week measurement.
Authors’ response
We are thankful to the Reviewer 1 for pointing this out. We agree with the Reviewer that it is indeed very interesting that PBR is not decreased in all timepoints for all participants. Although it is interesting, it is not surprising. To our knowledge, in clinical studies it is unlikely that all participants develop the same “response” or “size of effect” to a certain intervention. This is the reason why personalized medicine and “N-of-1” trials are increasingly being discussed in literature [Schork, Nicholas J. “Personalized medicine: Time for one-person trials.” Nature vol. 520,7549 (2015): 609-11. doi:10.1038/520609a]. Our study is not an exception and, admittedly, we expected (before the initiation of the study) that cART will not affect EG integrity in the same way among all participants as shown in Figure 1. This why we chose to present our primary outcome with a slope chart and not a box plot. Nonetheless, statistical analysis demonstrated that PBR was reduced in most cases and, therefore, our primary outcome was statistically significant.
However, we agree with the Reviewer 1 that it is worth explaining if these participants differed somehow compared to those who demonstrated a decrease in PBR compared to the baseline measurement. Therefore, we compared the baseline characteristics (sex, age, smoking status, body weight and cART regimen) of those whom PBR increased at 24 (±4) weeks and at 48 (±4) weeks compared to the baseline, and there was no statistically significant difference between the two groups in each timepoint respectively. Regarding this comment, we updated the Figure 1 legend as follows:
Page: 8, Lines: 242-249: “Figure 1. The slope chart shows the PBR5-25 (in μm) of each individual participant at baseline, at 24 (±4) weeks and at 48 (±4) weeks after initiation of antiretroviral treatment. PBR5-25 decreased in the majority of the participants in both timepoints, compared to baseline [65.5% and 69.5%, for 24 (±4) and 48 (±4) weeks, respectively]. Sex, age, antiretroviral treatment, body weight and smoking status did not differ significantly between the two groups (i.e., those whom PBR5-25 increased and those whom PBR5-25 decreased compared to the baseline) in both timepoints. PBR5-25: Permeability boundary region in sublingual microvessels with dimeter between 5 and 25 μm.”
Comment 5
There is a decrease in PBR. Is this biologically or clinically meaningful? What level of decrease is associated with a marker of clinical improvement?
Authors’ response
Thank you for this insightful comment. As already stated throughout our manuscript, EG degradation has been associated with several pathological conditions dictating that EG derangement can be either the cause or the result (or both) of the insult in all these conditions. Hence, measuring the magnitude of EG destruction (through measuring the PBR in our case) is both clinically and biologically meaningful, as depicts the damage on a microvascular level. Although the role of glycocalyx has long been recognized, there are very limited data quantifying and correlating the magnitude of EG thickness with different outcomes (such a disease severity, recovery, mortality, morbidity etc.) in general population. Currently, there is no “cut off” for PBR measurements that could evoke certain clinical implications. With regards to this comment, we added the following in our Discussion:
Page: 15, Lines: 473-481: “Based on these data, EG derangement can either be the cause and/or the result of endothelial damage in these conditions. Hence, evaluating the magnitude of EG destruction (through measuring the PBR) is both clinically and biologically meaningful, as depicts the damage on a microvascular level. Although the role of glycocalyx in endothelial protection has long been acknowledged, there are very limited data demonstrating that the numerical quantification of EG impairment could correlate with outcomes such as the dis-ease severity, morbidity, mortality etc. in non-HIV population. Hence, there is currently no numerical “cut off” value for PBR that could evoke certain clinical decisions. Further studies are warranted to establish whether there is a role of such “cut-offs” in clinical practice.”
Comment 6
I would remove the sections and discussion about PBR measurement differences based on integrase versus protease inhibitor differences and men versus women. The study was never designed to examine these differences. It does not have statistical power to make any statements regarding these confounders.
Authors’ response
We thank the Reviewer 1 for this comment. Although the study was not designed to address subgroup differences in regard to PBR trajectory, we believe that these results of exploratory character provide valuable information and should not be totally omitted in the study results. In any event, studies are not usually powered for secondary outcomes, such as these. Moreover, statistically significant differences found in the context of smaller subgroups further support the magnitude of detected changes. Naturally, there is a risk of a type II error when interpreting non-significant results in small groups. We have already acknowledged this limitation in the last paragraph of the discussion section. Please also refer to our response in Comment #8.
Comment 7
It is surprising that PBR is lower among those with higher as compared to lower virus level at baseline. If virus is driving EG derangement, I would have expected the opposite. How do investigators explain this?
Authors’ response
Thank you for pointing this out. Indeed, we (as well) expected the opposite result from a pathophysiological point-of-view. However, we have already given a possible explanation in our Discussion section (page: 15, lines: 456-465: “This result underlines that not only the presence of the virus is important for EG destruction, but also that the degree of EG impairment is proportional to the circulating viral loads. In contrast to this study, we found that PLWH with the lowest levels of HIV viremia experienced the greatest PBR improvement over time after treatment initiation, showing that, within the context of HIV infection, the insult to EG may not be directly mediated from the virus itself, but from the associated activation of the immune system. Although this finding was statistically significant, it should be highlighted that the number of PLWH with the lowest (<1,000 copies/ml) and highest (>500,000 copies/ml) initial viral loads was very small, and therefore a cautious interpretation of this result is warranted.”). As we already state in this paragraph, it should be emphasized that this result should be interpreted with caution, as the number of these subgroups were small.
Comment 8
The study likely does not statistical power to detect differences among smokers versus non-smokiers. Was the smoking status static over the 48 weeks? In other words, those classified as smokers continued to smoke over the 48 weeks and similarily for the non-smokers and ex-smokers?
Authors’ response
Thank you for this constructive comment. PBR changes according to smoking status was one of our secondary outcomes. The vast majority of clinical trials are powered for their primary outcome, as we did in our study. Being powered for all the outcomes (which is extremely rare) comes at the expense of a much larger sample size. We quote the following from a recent paper addressing the basic principles of sample size calculation: “At the outset, primary objectives (descriptive/analytical) and primary outcome measure (mean/proportion/rates) should be defined. Often there is a primary research question that the researcher wants to investigate. It is important to choose a primary outcome and lock that for the study. The minimum difference that investigator wants to detect between the groups makes the effect size for the sample size calculation…. In literature, there can be several outcomes for each study design. It is the responsibility of the researcher to find out the primary outcome of the study. Mostly sample size is estimated based on the primary outcome.” [Das S, Mitra K, Mandal M. Sample size calculation: Basic principles. Indian J Anaesth. 2016 Sep;60(9):652-656]. As already pointed out in our previous response to Comment #6, there is always a risk of a type II error when interpreting non-significant results in small groups (here, for the groups of non-smokers and ex-smokers), and we have already acknowledged this limitation in the last paragraph of the discussion section. Regarding the second part of this comment, smoking status remained unchanged over the course of the study for all participants. We agree with the Reviewer 1 that this should be better clarified in our manuscript. Therefore, we added the following into our Results section:
Page: 12, Lines: 315-317: “Although we based this subgroup analysis on the baseline smoking status, it should be mentioned that smoking status remained unchanged for all participants during the study period.”

Reviewer 2 Report
Fragkou et al. assessed EG in PLWH at three different time points during early stage of cART by measuring the PBR. This is an attractive non-invasive method and the measurements have been shown to be associated with heart diseases. HIV viral load and several biomarkers were also measured during the study. Data were analyzed as a whole or in different subgroups based on the treatments, sex, nadir CD4+ T cell count, HIV viral load, biomarkers, smoking status, illicit drug use and weight gain. Authors found EG thickness progressively improved one year after cART in PLWH. However, larger sample size is needed to assess the correlations with the variables.
Major concerns:
1. There is no age-matched normal/healthy HIV- individuals included in this study for comparison. It is important to know the true healthy baseline PBR and the “high” PBR that was shown to be associated with the heart diseases. Authors have listed some references, but the actual numbers should be quoted in the manuscript.
2. The definition of the “baseline” is confusing. Many places were listed as “before cART initiation”, but most figures listed as “0-6 weeks” or “6 weeks”. Please keep all figures, tables, and text consistent with the definition.
3. Figure 1 needs to be improved by providing the % of patients had decreased/increased PBR at 24 h and 48 h time points compared to the baseline.
Minor concerns:
1. Did authors observe any changes in PBR in the 13.6% PLWH patients with virological failure on cART? Can authors explain the reasons of the virological failure in this study?
2. Explain the orange dots in Figures 3 and 5.
3. How would you propose to use it as “possible therapeutic target” (Line 461)? Should clinicians routinely perform this test in PLWH population after certain years of cART treatment? Or do authors have other ideas to use it as a “therapeutic target”?
Author Response
Dear Editor and Reviewers,
Please find attached the revised manuscript entitled “Endothelial glycocalyx integrity in treatment-naïve people living with HIV before and one year after antiretroviral treatment initiation.” (MS ID#: viruses-2411319). On behalf of all co-authors, we greatly appreciate both Reviewers’ comments and wish to thank them both for their time and valuable input into our manuscript. In this revised submission we have addressed all comments and suggestions made by the Reviewers. Our point-by-point responses to each comment are provided below. The page and line numbers refer to the clean version of the manuscript.
REVIEWER #2
Comment 1
Fragkou et al. assessed EG in PLWH at three different time points during early stage of cART by measuring the PBR. This is an attractive non-invasive method and the measurements have been shown to be associated with heart diseases. HIV viral load and several biomarkers were also measured during the study. Data were analyzed as a whole or in different subgroups based on the treatments, sex, nadir CD4+ T cell count, HIV viral load, biomarkers, smoking status, illicit drug use and weight gain. Authors found EG thickness progressively improved one year after cART in PLWH. However, larger sample size is needed to assess the correlations with the variables.
Authors’ response
We are thankful to Reviewer 2 for these comments. We agree that subgroup analyses, which mainly were of an exploratory character, were based on small sample sizes, and larger cohorts are warranted to investigate these differences in future studies. However, we believe some interesting findings can be drawn by these preliminary results, and for this reason we decided to include them in our manuscript.
Major concerns:
Comment 2
There is no age-matched normal/healthy HIV- individuals included in this study for comparison. It is important to know the true healthy baseline PBR and the “high” PBR that was shown to be associated with the heart diseases. Authors have listed some references, but the actual numbers should be quoted in the manuscript.
Authors’ response
We appreciate this insightful point raised by the Reviewer 2. In this study we have already included healthy PLWH with not known cardiovascular disease. However, we agree that comparisons of our main outcome with a control group from general population would be very useful, as it would depict differences of PBR between PLWH and general population, that show the effect of infection on EG. In other words, a control group could show the “extent” of EG derangement in PLWH before cART initiation, compared to general population, and whether cART initiation improved EG thickness to such an extent that it returned to “normal”. Therefore, we searched the literature to find previous reports on PBR measurements in general population; interestingly, we found that although PBR is reduced one year after cART initiation, it does not reach the values of general population. Hence we added the following paragraph in our Discussion:
Page: 13, Lines: 369-382: “In our study we found that EG improves significantly within the first year of cART initiation in PLWH. However, it is very important to clarify whether this reduction represents a return to a “normal” EG integrity (or to PBR5-25 values that are comparable to these of general population) or EG derangement persists even after the initiation of cART initiation and the control of HIV viremia. Previous studies conducted in our center, have shown that PBR5-25 values in healthy adult controls (slightly older than our participants) ranges between 1.77 and 1.78 μm [29,30]. Similarly, a small observation-al study of healthy young adults measuring PBR5-25 before and after exercise, reported a baseline PBR5-25 of 1.86 μm [31]. Compared to the results of PBR5-25 at 48 ±4 weeks (1.93 μm) in our cohort, these values show that EG integrity is not completely restored among PLWH after one year of cART. Hence, cART does improve the microvascular effects of HIV infection but only partially (at least within the first year of treatment), and other factors of endothelial damage (such as chronic immune activation) should also be taken under con-sideration in HIV infection.”
Comment 3
The definition of the “baseline” is confusing. Many places were listed as “before cART initiation”, but most figures listed as “0-6 weeks” or “6 weeks”. Please keep all figures, tables, and text consistent with the definition.
Authors’ response
We thank the reviewer for pointing that out. We have kept the term “baseline” and we changed the other terms accordingly, both in the manuscript and the figures.
Comment 4
Figure 1 needs to be improved by providing the % of patients had decreased/increased PBR at 24 h and 48 h time points compared to the baseline.
Authors’ response
Thank you very much for this comment. For this comment please also refer to our response to Comment 4 by the Reviewer 1. We agree with the Reviewer 2 that some of the PBR values increase compared to the baseline and providing the respective percentages will be useful for the readership of the manuscript. Furthermore, it is worth explaining if these participants differed somehow compared to those who demonstrated a decrease in PBR compared to the baseline measurement. Therefore, we compared the baseline characteristics (sex, age, smoking status, body weight and cART regimen) of those whom PBR increased at 24 (±4) weeks and at 48 (±4) weeks compared to the baseline, and there was no statistically significant difference between the two groups in each timepoint respectively. Regarding this comment, we updated the Figure 1 legend in our revised manuscript as follows:
Page: 8, Lines: 242-249: “Figure 1. The slope chart shows the PBR5-25 (in μm) of each individual participant at baseline, at 24 (±4) weeks and at 48 (±4) weeks after initiation of antiretroviral treatment. PBR5-25 decreased in the majority of the participants in both timepoints, compared to baseline [65.5% and 69.5%, for 24 (±4) and 48 (±4) weeks, respectively]. Sex, age, antiretroviral treatment, body weight and smoking status did not differ significantly between the two groups (i.e., those whom PBR5-25 increased and those whom PBR5-25 decreased compared to the baseline) in both timepoints. PBR5-25: Permeability boundary region in sublingual microvessels with dimeter between 5 and 25 μm.”
Minor concerns:
Comment 5
Did authors observe any changes in PBR in the 13.6% PLWH patients with virological failure on cART? Can authors explain the reasons of the virological failure in this study?
Authors’ response
We are grateful to the Reviewer 2 for this insightful comment. We added the following into our revised manuscript:
Page: 7, Lines: 214-215: “Nine (13.6%) out of 66 PLWH had virological failure with detectable viral load at visit 3, potentially attributed to poor compliance with their daily medication.”
Page: 11, Lines: 293-298: “Finally, we compared the PBR5-25 at 48 (±4) weeks and the PBR5-25 difference between the last and the first visit among PLWH with and without virological failure to test whether incomplete compliance with treatment could affect the EG integrity; we found no statistically significant difference in PBR5-25 and PBR5-25 difference between these two groups (p= 0.839 and p=0,411, respectively).”
Comment 6
Explain the orange dots in Figures 3 and 5.
Authors’ response
We thank the reviewer for bringing this in our attention. Orange dots are supposed to depict outliers, but since it could cause confusion, we omitted them in our revised manuscript.
Comment 7
How would you propose to use it as “possible therapeutic target” (Line 461)? Should clinicians routinely perform this test in PLWH population after certain years of cART treatment? Or do authors have other ideas to use it as a “therapeutic target”?
Authors’ response
We highly appreciate this very constructive comment by the Reviewer 2. With regards to this point we added the following into our Discussion section:
Page: 15-16, Lines: 484-494: “Various approaches for glycocalyx regeneration have been pursued in the context of reversing endothelium damage leading to atherosclerotic plaque formation. Most commonly, atherosclerotic plaque stabilizers (e.g., sulodexide and rosuvastatin), antidiabetic drugs (metformin), anti-inflammatory therapies, anticoagulants and dietary supplements have been investigated, both in animal and human studies, with some early but promising results [41,50]. Glycocalyx repeated measurements could also be employed to assess response to existing treatments for CVD. However, there still are barriers in the wide-spread use of glycocalyx measurement, that are related to assay precision, clinical interpretation of findings, and lack of homogeneity between different assessment methods [51]. Hence, there is still a long way (in term of research) before EG integrity evaluation could be established as a routine test for CVD assessment in PLWH.”

Round 2
Reviewer 2 Report
The authors have addressed my comments and made sufficient changes.